

# Storm characteristics influence nitrogen removal in an urban estuarine environment

Anne Margaret H. Smiley[1], Suzanne P. Thompson[2], Nathan Hall[2], Michael F. Piehler[1,2,3]

[1]Department of Earth, Marine, and Environmental Sciences, University of North Carolina, Chapel Hill, 27514, USA
[2]UNC Institute of Marine Sciences, Morehead City, 28557, USA
[3]UNC Institute for the Environment, Chapel Hill, 27517, USA

*Correspondence to*: Anne Margaret H. Smiley (ahsmiley@live.unc.edu)

**Abstract.** Sustaining water quality is an important component of coastal resilience. Floodwaters deliver reactive nitrogen ($NO_x$) to sensitive aquatic systems and can diminish water quality. Coastal habitats in flooded areas can be effective at

removing reactive nitrogen through denitrification (DNF). However, less is known about this biogeochemical process in urbanized environments. This study assessed the nitrogen removal capabilities of flooded habitats along an urban estuarine coastline in the upper Neuse River Estuary (NRE), NC, USA under two nitrate concentrations (16.8 µM and 52.3 µM $NO_x$, respectively). We also determined how storm characteristics (e.g., precipitation and wind) affect water column $NO_x$ concentrations and consequently DNF by flooded habitats. Continuous flow-through sediment core incubation experiments

quantified gas and nutrient fluxes across the sediment-water interface in marsh, swamp forest, undeveloped open space, stormwater pond, and shallow subtidal sediments. All habitats exhibited net DNF. Additionally, all habitats increased DNF rates under elevated nitrate conditions compared to low nitrate. Structured habitats with high sediment organic matter had higher nitrogen removal capacity than unstructured, low sediment organic matter habitats. High precipitation-high wind storm events produced concentrations significantly lower than other types of storms (e.g., low precipitation-high wind, high wind-

low precipitation, low wind-low precipitation), which likely results in relatively low DNF rates by flooded habitats and low removal percentages of total dissolved nitrogen loads. These results demonstrate the importance of natural systems to water quality in urbanized coastal areas subject to flooding.

## 1 Introduction

Tropical cyclones often cause extensive flooding that can harm ecosystems, damage infrastructure, and disrupt the lives of

coastal residents. There is evidence to suggest that anthropogenic climate change has produced conditions (e.g., warmer sea surface temperatures, increased atmospheric moisture) that make these high magnitude events more likely (Knutson et al., 2013; Min et al., 2011). Since the mid-1990's there has been an observed shift in storm activity in the United States where tropical cyclones have become slower and rainier, and result in catastrophic flooding at higher frequencies than historical averages (Easterling et al., 2017; Kossin, 2018; Kunkel et al., 2010). As climate change continues, some models predict an

increase in the most intense storms and up to a 20 % increase in precipitation rates by 2100 (Knutson et al., 2010).



Floodwaters introduce allochthonous materials, including nutrients, to downstream receiving waters. Storm-related upstream discharge typically contains high concentrations of inorganic nitrogen and organic carbon, that constitute up to 80 % of annual loads into receiving waterbodies (Paerl et al., 2020). Estuaries are often nitrogen limited and sensitive to sudden influxes of

reactive nitrogen (Howarth & Marino, 2006), therefore floodwaters can trigger water quality degradation by fueling algal blooms (Nixon, 1995) that can disrupt aquatic ecosystems by outcompeting other vegetated habitats for sunlight and nutrients (Wasson et al., 2017). One of the largest cyanobacterial blooms in the Okeechobee region has been attributed to Hurricane Irma in 2017 (Hampel et al., 2019) and models showed a strong biological response in Apalachicola Bay following Hurricane Michael in 2018 (D'Sa et al., 2019). Remineralization of algal biomass and terrigenous organic matter by heterotrophic bacteria

depletes oxygen in the water column, which can affect health of aquatic organisms and the ecosystem overall  (Diaz & Roseberg, 1995).

Watershed urbanization has been shown to exacerbate water quality degradation in tropical, subtropical, and temperate coastal regions  by interfering with hydrologic, geomorphic, and biogeochemical processes (Bowen & Valiela, 2001; Gold et al., 2019,

2021; Lee et al., 2006; Ortiz-Zayas et al., 2006). Population growth has led to increased point source nutrient loading via wastewater effluent into receiving waterways (Carey & Migliaccio, 2009; Naden et al., 2016). Furthermore, impervious surfaces and stormwater pipes can streamline flow paths and enhance the export of non-point source anthropogenic nitrogen (Bernhardt et al., 2008). Agricultural landscapes can also deliver nutrients to receiving waterways. In some regions, high densities of agricultural operations substantially increase nutrient concentrations from nitrogen-based fertilizer and animal

waste (Duda, 1982; Dupas et al., 2015).

Some natural habitats have been shown to be effective at removing terrigenous and anthropogenic nitrogen through a series of biogeochemical reactions (Groffman & Crawford, 2003; Pérez-Villalona et al., 2015; Piehler & Smyth, 2011; Reisinger et al., 2016; Rosenzweig et al., 2018). Denitrification (DNF) is a process by which sediment microbes convert bioavailable forms of

nitrogen (nitrate and nitrite) to dinitrogen gas ($N_2$) under anaerobic conditions using carbon as an energy source. DNF is an important process by which reactive nitrogen is naturally and permanently removed from a system. It can be an effective strategy for maintaining water quality during flood conditions that favor DNF, namely, elevated dissolved nitrate and carbon, and anoxia (Adame et al., 2019; Velinsky et al., 2017). Much work has been done to understand DNF by natural habitats, such as emergent wetlands and oyster reefs (Ensign et al., 2008; Grabowski et al., 2012; Onorevole et al., 2018; Piehler & Smyth,

2011; Velinsky et al., 2017), but much less is known about nitrogen processing by urban landscapes, such as stormwater ponds and lawns/undeveloped open space (UOS), despite their being prolific in developed settings. The primary objective of this study is to quantify nitrogen removal by denitrification in flood-prone habitats, both natural and human influenced, including marsh, forested wetland, stormwater pond, undeveloped open space, and shallow subtidal sediments under varied nutrient conditions in Neuse River Estuary (NRE), North Carolina (NC).






Storms exhibit unique characteristics which can affect water chemistry differently (Davis et al., 2004; Mallin et al., 2002; Wetz & Paerl, 2008). Some storms produce elevated nutrient concentrations. Studies have shown that sustained winds at high speeds can increase nutrient concentrations by mixing stratified waters and resuspending sediments (Goodrich et al., 1987; Miller et al., 2006; Wengrove et al., 2015). Storms characterized by high precipitation can dilute the nutrients in the water column

(Minaudo et al., 2019). Paerl et al. (2020) described the Neuse River Estuary (NRE), in eastern North Carolina, as either a "processor" under relatively lower discharge periods where nutrients are able to be partially processed, or a "pipeline" during high discharge periods where nutrients are delivered to the Albemarle-Pamlico Sound with little processing in the NRE. Therefore, the nitrogen removal capacity of flooded landscapes via DNF is likely influenced by water quality produced during varied storm conditions as well as contact time of floodwaters prior to export from the system. A secondary objective of this

work is to compare nitrogen loads during multiple storm types to projected nitrogen removal rates by predominant landscape habitats including inundated marshes, forested wetland, UOS, and subtidal sediments.

As urban landscapes expand, resulting in losses of natural habitats and wetlands (Aguilera et al., 2020) concomitant with increased anthropogenic nutrient loads, it is essential that we understand the role that both natural and human influenced

landscapes play in removing reactive nitrogen. Additionally, assessing how these habitats perform under a range of nutrient conditions will enable us to estimate landscape scale nitrogen removal capacities during different types of storms. These data will improve our understanding of estuarine nutrient budgets along urbanized coastlines in a new regime of tropical cyclone activity and inform strategic coastal development that conserves ecosystem services to maximize benefits for coastal residents.

## 2 Methods

### 2.1 Approach

This study combines laboratory, computational modeling, and geographic information systems (GIS) methods to understand landscape-scale DNF capacity during different types of storms. Storm types were defined based on precipitation and wind characteristics. $NO_x$ concentrations, $NO_x$ loads, and TDN loads during those storm events were modeled using weighted regressions. Habitat-specific DNF rates under ambient and elevated nitrate conditions were determined through laboratory

experiments. These nitrate treatments represented low and high water column $NO_x$ concentrations that are likely associated with different types of storms. Nitrogen removal during these storms was estimated based on experimental DNF values and inundated area of each habitat treatment at maximum inundation. These results were used to draw conclusions about the influence of storm characteristics on water column $NO_x$ concentrations, and consequently, biogeochemical processes in flooded landscapes.





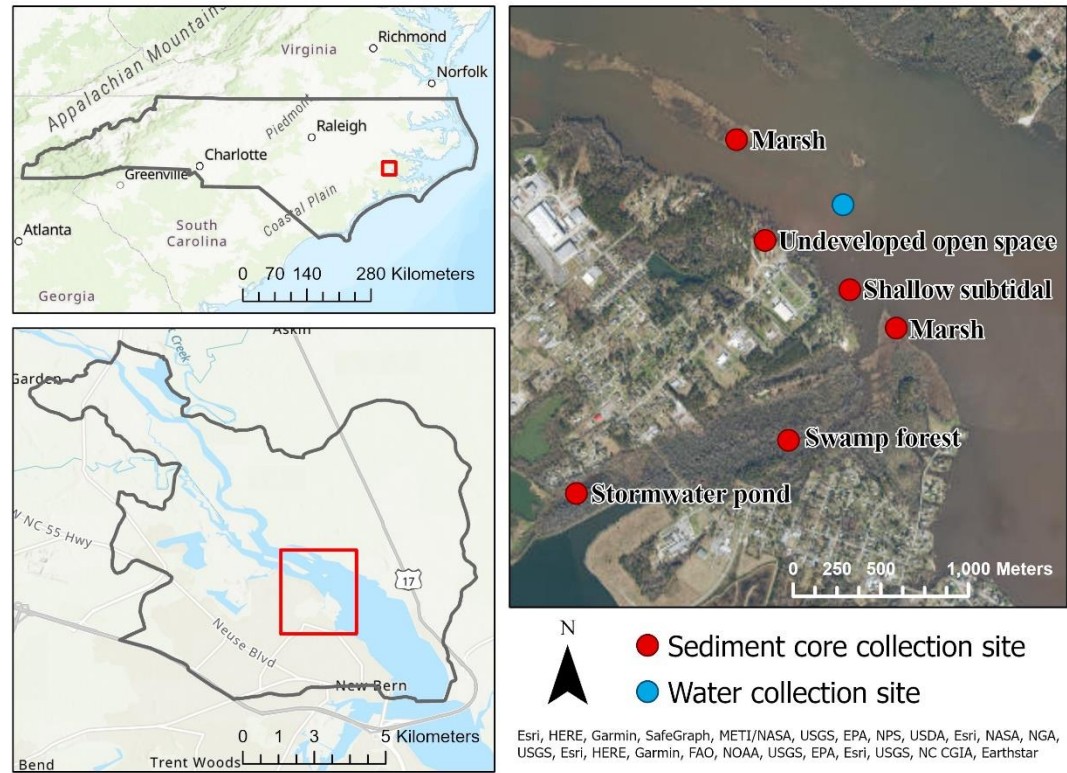

**Figure 1. Site map of New Bern, NC in the upper Neuse River Estuary and sampling locations for sediment core and water collection.**

## 2.2 Site description

The upper reaches of the Neuse River Estuary (NRE), North Carolina, USA are influenced by both riverine and coastal hydrologic processes, making them prone to multiple forms of flooding (e.g., fluvial, pluvial, and coastal storm surge). This region also includes both highly urbanized areas and natural ecosystems, affording the opportunity to assess anthropogenic impacts on ecosystem functioning in the context of a built environment. NRE is nutrient sensitive (Boyer et al., 1994; Pinckney et al.; Rudek et al., 1991); primary production is primarily nitrogen limited and episodic loading events can result in water quality degradation. With headwaters at the urban center of Raleigh and several smaller cities distributed along the river and throughout the watershed, the NRE receives inputs from a 16,000 km² drainage basin (Christian et al., 1991). Extensive agricultural use paired with rapid urbanization within the watershed makes NRE and similar locations susceptible to water quality degradation during major flood events.



## 2.3 Storm classification and water quality characteristics

### 2.3.1 Storm types based on wind and precipitation

Paerl et al. (2018) categorized tropical cyclones that made landfall in North Carolina between 1996 and 2016, based on river
discharge at Fort Barnwell and wind speeds at Cape Lookout, NC (NOAA National Data Buoy Center Station CLKN7). The same criteria were used to categorize storms between 2017 and 2019. Storms that resulted in a 7-day mean Neuse River discharge above the 90th percentile of weekly averages (191 m³ s⁻¹) were designated "high precipitation" (HP) events. Those that exhibited a maximum hourly average wind speed above the 90th percentile (14.1 m s⁻¹) between the 12 hours prior to and 24 hours after landfall, were considered "high wind" (HW) events. Storms that produced riverine discharges or wind speeds
below these thresholds were considered "low precipitation" (LP) and "low wind" (LW) events, respectively. Storm types were assigned based on both precipitation and wind classifications (Table 1). For example, Hurricane Florence produced both high precipitation and high wind conditions and is therefore labelled as a HP-HW storm. Furthermore, storms that were considered both LP-LW were thought of as "baseline storm" conditions.

**Table 1. Summary matrix of named storms that made landfall on North Carolina's coast between 1996 and 2019 categorized by storm type derived from Neuse River discharge and average wind speeds. Red text indicates storms with available floodplain footprints that were assessed for nitrogen removal.**

|  | High precipitation (HP) | | Low precipitation (LP) | |
|---|---|---|---|---|
| **High wind (HW)** | Fran (1996) Josephine (1996) Dennis (1999) Floyd (1999) Irene (1999) Gordon (2000) | Ernesto (2006) Irene (2011) Joaquin (2015) Matthew (2016) Florence (2018) | Arthur (1996) Bertha (1996) Bonnie (1998) Earl (1998) Helene (2000) Gustav (2002) Isabel (2003) | Ophelia (2005) Barry (2007) Earl (2010) Beryl (2012) Andrea (2013) Arthur (2014) Hermine (2016) |
| **Low wind (LW)** | Charley (2004) Nicole (2010) Ana (2015) Dorian (2019) | | Danny (1997) Allison (2001) Alex (2004) Bonnie (2004) | Gaston (2004) Gabrielle (2007) Christobol (2008) Hanna (2008) |



### 2.3.2 Water quality characteristics during multiple storms

This study compared water quality characteristics during multiple tropical cyclones that affected the North Carolina coast. Average $NO_x$ concentrations, $NO_x$ loads, and TDN loads on the day of maximum riverine discharge were estimated using Weighted Regressions on Time Discharge and Season (WRTDS) (Hirsch et al., 2010) as described in Paerl et al. (2018) with the following exception. The half window width for the flow term in the model was reduced from the default of 2 ln(flow) increments down to 1 ln(flow) increment to provide greater resolution of flow impacts on concentration and fluxes. This was necessary to capture the observed strong dilution effect of nitrate during extreme, storm-induced flood events. Average concentrations and loads were compared across storm types.

### 2.4 Nitrogen flux experiment

### 2.4.1 Sample collection

Sampling for nitrogen flux experiments occurred in October of 2020, timed to capture typical environmental conditions during hurricane season. Sediment cores (6.4 cm diameter with a height of approximately 17 cm) were collected in triplicate from habitats subject to storm flooding including subtidal sediments, stormwater pond, UOS, swamp forest and two marshes—one upstream and one downstream of the outfall from the City of New Bern's wastewater treatment facility (Figure 1). The two marshes did not exhibit significant differences between mean flux rates and were, therefore, treated as a single habitat treatment.

### 2.4.2 Dissolved gas and nutrient fluxes across the sediment-water interface

The sediment cores and water were taken to UNC-CH's Institute of Marine Sciences in Morehead City, NC to conduct dissolved gas and soluble nitrogen flux experiments using methods described by Piehler and Smyth (2011). Sediments and site water were incubated in a temperature-controlled chamber (Bally Inc.) set to *in situ* water temperature (19 ℃) in a continuous flow-through system of water collected from the Neuse River between the two marshes (feedwater). The ambient $NO_x$ concentration of the feedwater was 16.8 µM. A peristaltic pump was used to pull feedwater through the sediment cores at a rate of 0.6 L h$^{-1}$, equating to a turnover time of 5-6 hours for water over the sediment in each core. After an overnight equilibration period, water samples were collected from each sediment core outflow for three timepoints, each 5 hours apart. Water pumped directly from feedwater bins was collected at each timepoint to assess inflow concentrations of dissolved gases and nutrients. After collecting the third timepoint, the feedwater was enriched with sodium nitrate to a concentration of 52.3 µM to simulate a relatively high nitrogen scenario. Following a second overnight equilibration period under nitrate enriched conditions, three more timepoints were collected 5 hours apart. These two nitrogen treatments are referred to as "low nitrate" (16.8 µM ambient $NO_x$ concentration) and "high nitrate" (52.3 µM enriched $NO_x$ concentration).



Directly following each water collection (6 total), a membrane inlet mass spectrometer (MIMS; Bay Instruments, Easton, MD) was used to analyze ratios of concentrations of dissolved gases, including $N_2 : Ar$ and $O_2 : Ar$, within water samples pulled from cores as well as those collected directly from the feedwater. These measurements were used to calculate net DNF rates and sediment oxygen demand (SOD). At timepoints 2 and 5, additional 50 mL water samples were collected to measure nutrient fluxes based on core inflow and outflow concentrations. Samples were filtered through 0.7 µm Whatman GF/F filters and

stored at -18 ˚C prior to analysis with a Lachat Quick-chem 8000 (Lachat Instruments, Milwaukee, WI, USA). Nutrient analytes included dissolved inorganic nitrogen species (DIN): nitrate + nitrite ($NO_x$) and ammonium ($NH_4^+$) as well as total dissolved nitrogen (TDN) allowing calculation of dissolved organic nitrogen (DON) by difference. At the end of the experiment, water was drained from the cores and sediment samples were collected from the top 2 cm to determine percent sediment organic material (SOM) based on loss on ignition (Byers et al., 1978; Smyth et al., 2015).

### 2.5 Spatial data acquisition

#### 2.5.1 Habitat treatments

Distributions and total surface area of sampled habitats were determined using a variety of spatial datasets. Marshes, swamp forests, and UOS were delineated using the National Land Cover Dataset (NLCD), a 30m raster dataset obtained from remotely sensed Landsat imagery. Land cover classified as emergent and forested wetlands were considered marsh and swamp forest,

respectively. Area of UOS was calculated by combining the herbaceous classification and weighted estimates from the various development categories (e.g., open space, low-high intensity). Pixels considered "Developed, open space" in the NLCD are defined as those comprised of less than 20 percent constructed surfaces. The remaining 80 % of the pixel area was considered UOS, colloquially referred to as lawns and grasses. Low, medium, and high intensity developed pixels were considered 51 %, 21 %, and 0 % UOS, respectively. NLCD datasets have been updated roughly in 2 to 3-year intervals. This work references

datasets from multiple years, including 2004, 2016, and 2019.

Shallow subtidal sediments were identified using NOAA's Continuously Updated Digital Elevation Model (CUDEM) and were defined as those within 1 meter of the surface. Stormwater infrastructure data were obtained from the City of New Bern and include managed stormwater ponds. ArcGIS Pro 2.8.7 was used to extract NLCD and CUDEM data from 2 HUC12

watersheds in the upper NRE.

#### 2.5.2 Inundation extents for multiple storms

Flood footprints that delineated inundated landscapes for seven selected storms were generated from the Advanced Circulation (ADCIRC) model and acquired from the Coastal Emergency Risks Assessment website (https://cera.coastalrisk.live/). This analysis includes Hurricanes Charley (8/23/2004), Arthur (7/4/2014), Joaquin (10/12/2015), Hermine (9/2/2016), Matthew




(10/8/2016), Florence (9/26/2018), and Dorian (9/6/2019). Flood footprints for LP-LW storms were not available, so these types of baseline storm events were not considered.

## 2.6 Calculations and statistical analysis

### 2.6.1 Dissolved gas and nutrient fluxes

Fluxes of nutrient and dissolved gases across the sediment water interface were calculated by multiplying the difference
between inflow and outflow concentrations by the peristaltic pump/flow rate and dividing by the surface area of the sediment core as in Eq. (1).

$$Flux = \frac{(C_{outflow} - C_{inflow}) \times F}{A} \tag{1}$$

Denitrification efficiency was calculated by dividing $N_2$-N fluxes by the total inorganic nitrogen flux out of the sediments,
then multiplying 100, following Eq. (2).

$$Denitrification\ efficiency = \frac{Flux_{(N_2-N)}}{Flux_{(N_2-N)} + Flux_{(NO_x-N)} + Flux_{(NH_4-N)}} \times 100 \tag{2}$$

Mean flux rates for dissolved gases and nutrients were compared using Kruskal Wallis and post hoc Dunn tests to identify differences across landscape treatments and between nutrient treatments. Linear regressions were performed to compare
variations in $N_2$-N fluxes to variations in SOD under ambient and nitrate enriched conditions. Additional linear regressions were used to compare variability in DNF to SOM under both low nitrate and high nitrate conditions. All statistical tests were done using R version 4.1.1 (R Core Team, 2011) and were considered significant when $p < 0.05$.

### 2.6.2 Nitrogen concentrations and loads during storms

Mean $NO_x$ concentrations, $NO_x$ loads, and TDN loads on the day of maximum riverine discharge were compared across storm
types were compared using Kruskal Wallis and post hoc Dunn tests. Differences between mean $NO_x$ concentrations were used to draw comparisons between experimental nitrate treatments to environmental $NO_x$ concentrations during different types of storms. Mean load values were compared to estimate nitrogen removal by flooded landscapes during multiple storms.

### 2.6.3 Nitrogen removal by the flooded landscape

Nitrogen removal was estimated for seven selected storms with available flood footprints. Tools in ArcGIS Pro were used to
extract land cover data from each storm's flood footprint within the two HUC12 watersheds in the upper NRE. The 2004 NLCD dataset was used to estimate inundated area during Hurricane Charley, the 2016 dataset was used to estimate inundated area for Hurricanes Arthur, Joaquin, Hermine, and Matthew, and the 2019 dataset was used for Hurricanes Florence and Dorian. For each habitat type, inundated surface areas and mean DNF rates obtained from the nitrogen flux experiments were



multiplied to estimate habitat-specific N removal rates, as in Eq. (3). Areas of shallow subtidal sediments were assumed to

have remained constant over this time range. Nitrogen removal by stormwater ponds was not considered in this analysis. Removal rates under both high and low nitrate conditions were calculated.

$$N\ removal\ rate = DNF\ rate\ x\ surface\ area \tag{3}$$

## 3 Results

### 3.1 Storm characteristics

HP-HW storms yielded a mean $NO_x$ concentration of 11.7 ± 2.50 µM on the day of maximum riverine discharge, which was lower than mean concentrations for the other three storm types (HP-LW: 25.2 ± 6.00; LP-HW: 29.3 ± 2.23; LP-LW: 24.5 ± 2.16; Figure 2). The low nitrate experimental treatment (16.8 µM) was considered more representative of the HP-HW events, while the high nitrate treatment (52.3 µM) was considered more analogous to the other three storm types. Mean $NO_x$ loads on

the day of maximum discharge during low precipitation storms were significantly lower than loads during high precipitation storms (Figure 2). TDN loads during HP-HW events were higher than low precipitation storms, and LP-LW events produced loads lower than high precipitation events. There were no significant differences in mean TDN load between HP-LW and LP-HW storms (Figure 2).

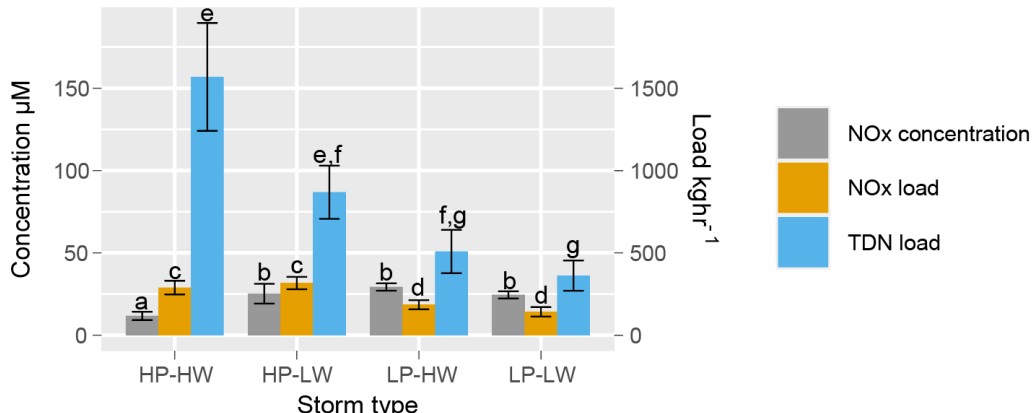


**Figure 2. Average $NO_x$ concentrations, $NO_x$ loads, and average TDN loads on the day of maximum river discharge for each storm type.**

### 3.2 Nitrogen fluxes across the sediment-water interface

Under the low nitrate conditions, all habitats exhibited net DNF (Figure 3a). $N_2$-N fluxes in marsh sediments were significantly

higher than shallow subtidal, swamp forest, and stormwater pond sediments (Figure 3a). Following nitrate enrichment, all





landscapes experienced a significant increase in DNF rates compared to respective flux rates under low nitrate conditions (Figure 3a). Under high nitrate conditions, marsh and stormwater pond cores produced significantly higher DNF rates than both UOS and subtidal sediments. Swamp forest cores also exhibited higher rates than the subtidal cores (Figure 3a).

$NO_x$ flux hovered near 0 µmol $m^{-2}$ $h^{-1}$ for each habitat under low nitrate conditions (Figure 3b), with no significant differences evident between means. Following the nitrate addition, each habitat exhibited a significant decrease in flux rates (Figure 3b). High nitrate $NO_x$ flux rates were negative for all habitat treatments, indicating $NO_x$ moving from the water column into the sediments; thus, each habitat acted as a $NO_x$ sink post-enrichment. $NO_x$ flux rates were not different between habitats. $NH_4^+$ fluxes were an order of magnitude lower than $N_2$ and $NO_x$ fluxes (Figure 3c). Some $NH_4^+$ fluxes were positive while
others were negative, although no significant differences across habitat or nitrate treatments were evident.

Average DNF efficiencies for all habitats and nitrate treatments were above 70 % (Figure 3d). Under low nitrate conditions, UOS was the most efficient habitat, significantly more efficient than marsh, stormwater pond, and shallow subtidal sediments. Following nutrient enrichment, marsh, stormwater pond, and shallow subtidal sediments showed a significant increase, and all
habitat treatments nearly reached 100 % DNF efficiency.

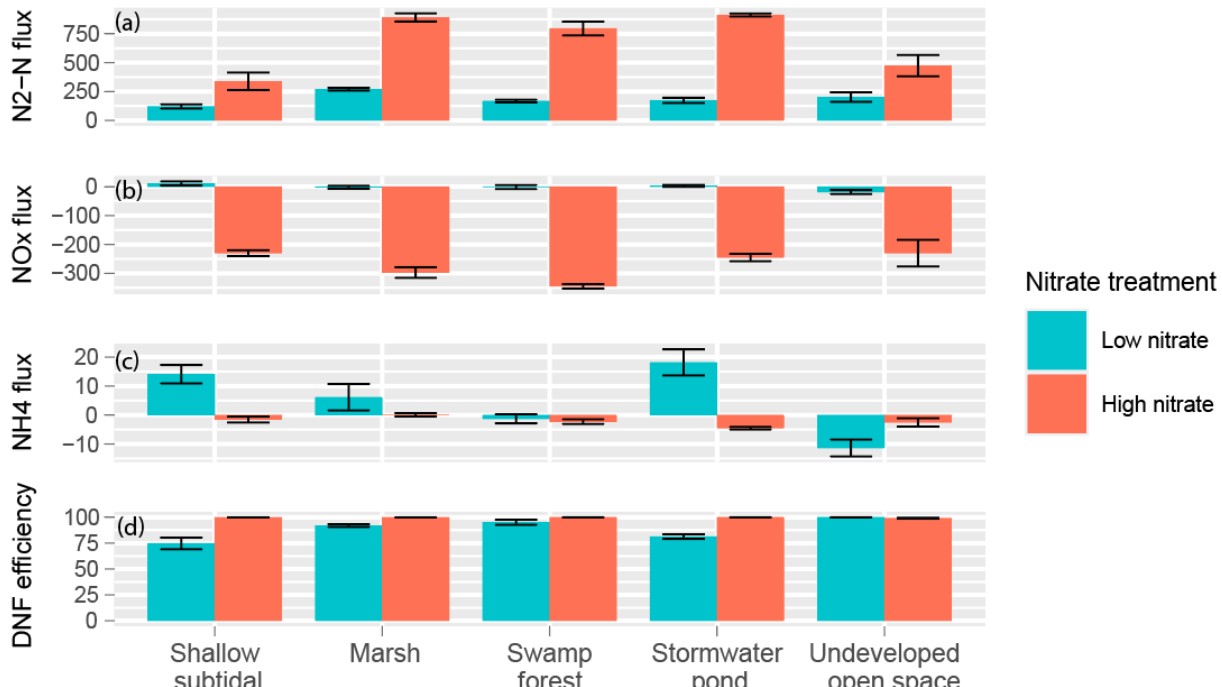

**Figure 3. Flux rates (µmol $m^{-2}h^{-1}$) across the sediment water interface for multiple nitrogen species, including: (A) $N_2$-N, (B) $NO_x$, and (C) $NH_4$, as well as (D) DNF efficiencies (%). Positive fluxes indicate movement from the sediments to the overlying water column.**




A linear regression analysis revealed that under low nitrate conditions, variability in SOD explains approximately 70 % of the variability in DNF (Figure 4). No significant relationship was evident between DNF and SOD under high nitrate conditions. There was a significant relationship between SOM and high nitrate DNF rates, where variability in SOM accounted for roughly 62 % of the variability in $N_2$-N flux.


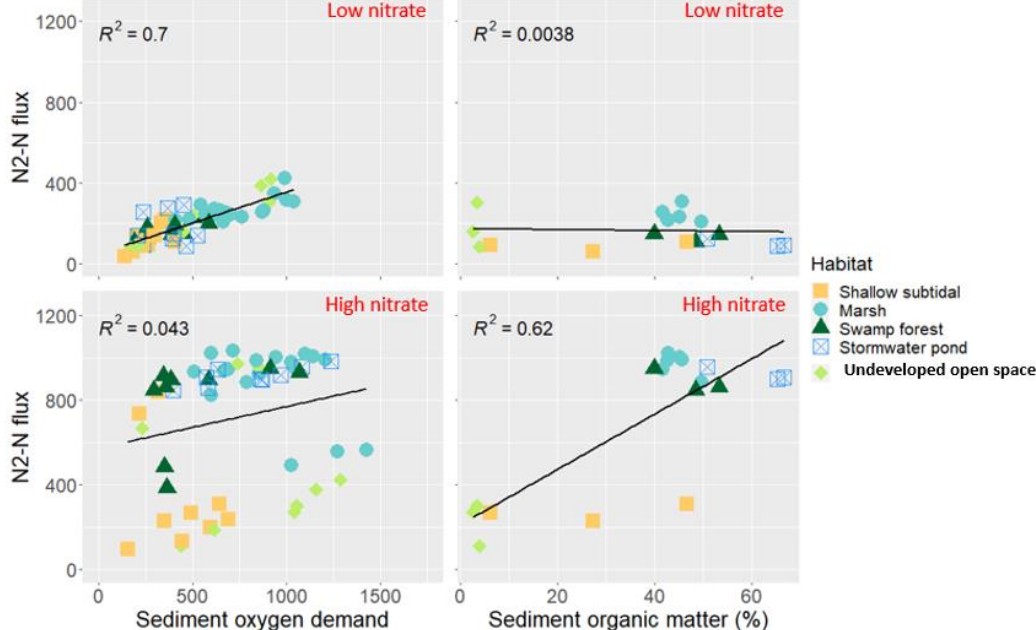

**Figure 4. Scatterplot and linear regression for the relationship between SOD and DNF under low (top left) and high (bottom left) nitrate conditions as well as SOM and DNF under low (top right) and high (bottom right) nitrate conditions.**

### 3.3 Nitrogen removal during storms

Nitrogen removal was calculated for seven named storms (of 3 precipitation/wind types, Table 1) with available flood footprints. This was done by multiplying habitat specific DNF rates across inundated surface areas for each habitat treatment using rates produced during both low and high nutrient treatments. Flood footprints were not available for any LP-LW storms, so these baseline storm events were not included in this analysis. Removal rates calculated using low nitrate DNF rates ranged between 15.3 and 65.5 kg N h$^{-1}$. High nitrate removal rates ranged between 58.4 and 257 kg N h$^{-1}$. Low nitrate removal rates

were considered more representative for HP-HW events and high nitrate removal rates were considered more representative for HP-LW and LP-HW events.



These removal rates were used to calculate the percent of TDN and $NO_x$ loads that were removed by habitats within the floodplains of the seven selected storms. Under low nitrate conditions, the percentage of TDN load removed ranged from 1.15

to 5.95; under high nitrate conditions, they ranged from 4.81 to 24.6. Regarding $NO_x$ loads, under low nitrate conditions, percent removed ranged from 5.71 to 21.6. Under high nitrate conditions, they ranged from 21.8 to 84.6.

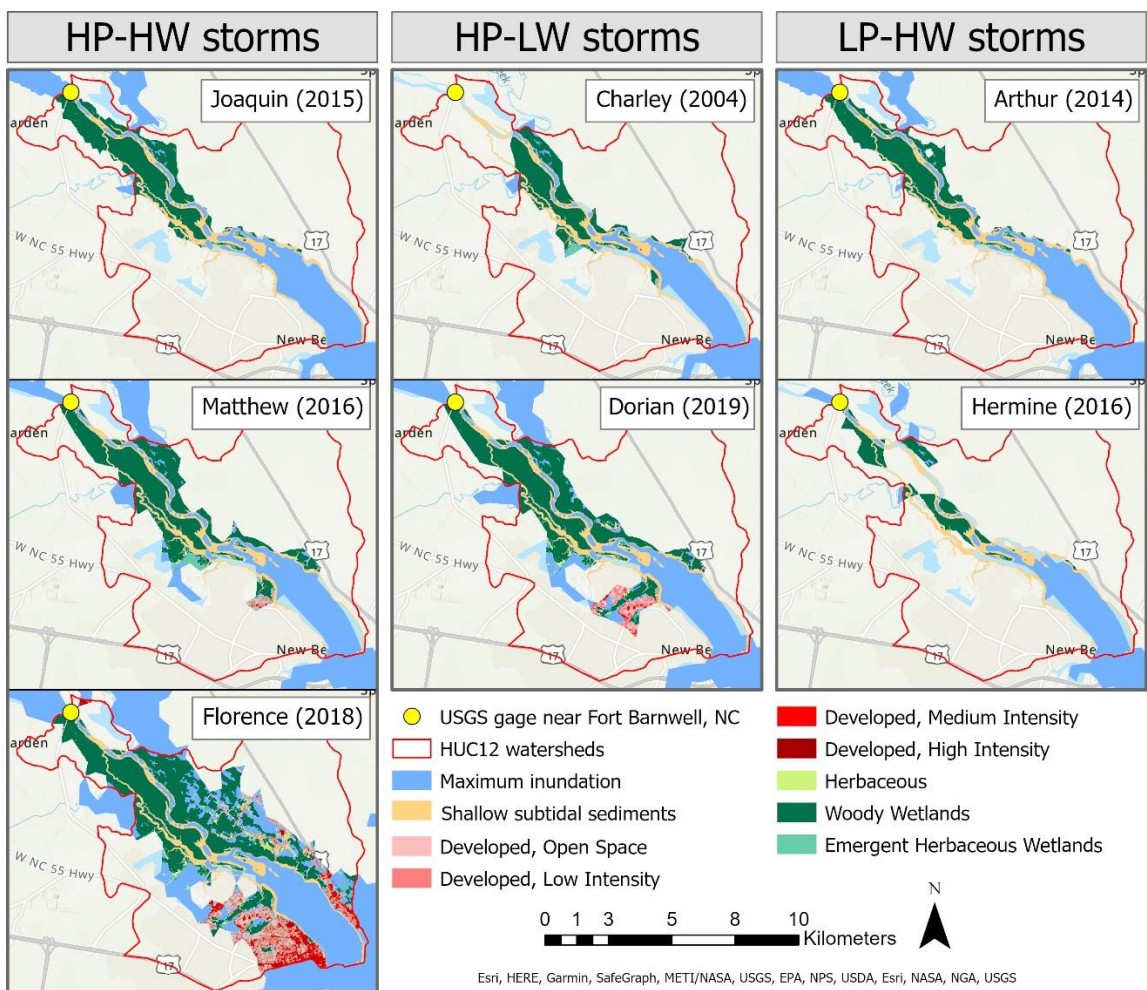

**Figure 5. NLCD land cover classifications within HUC12 watershed boundaries and floodplain footprints for multiple storms**
**affecting the Neuse River Estuary. Habitat treatments are derived from these land cover classifications.**

Floodplain footprints varied in size, with each storm inundating different proportions of each habitat (Figure 5). In each case, swamp forests were the most abundant inundated habitat in the floodplain, comprising between 44.9 and 66.2 % of the flooded habitat area. Their abundance paired with their relatively high DNF rates is reflected in their high contribution to

nitrogen removal overall (Figure 6). Swamp forests were likely responsible for removing between 51.6 and 70.1 % of





nitrogen removed

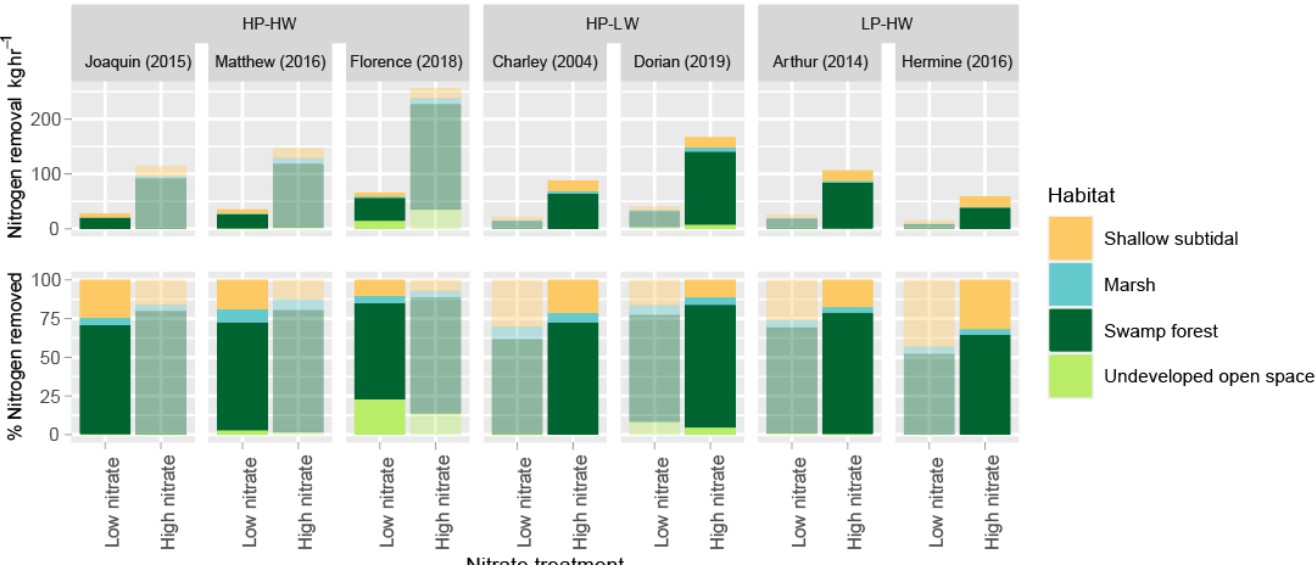

**Figure 6. Projected nitrogen removal rates (top) and percent contribution (bottom) by each habitat under low and high nitrate conditions during multiple storms. Higher color intensity indicates applicable nitrogen level based on storm type.**


via DNF by all habitats under low nitrate conditions and between 64.0 and 79.4 % under high nitrate conditions. Shallow subtidal sediments also consistently comprised a large proportion of flooded habitat area, between 12.4 and 52.2 %. Although DNF rates in this habitat are relatively low, their abundance led to large contributions to nitrogen removal during storms, between 10.1 and 43.3 % under low nitrate conditions and between 7.27 and 31.6 % under high nitrate conditions. Marshes
are relatively sparse in this region of the NRE that reliably contributed a small percentage of nitrogen removal. UOS also consistently made up a small portion of inundated landscapes, though the contribution of UOS seems to have increased in the most recent storms: Matthew, Florence, and Dorian (2.33, 13.0, and 6.57 % of the flooded habitat area, respectively).

Another consideration is the effect that $NO_x$ concentrations have on the relative contribution to overall nitrogen removal by
each habitat. Not all habitats respond the same to elevated nitrate conditions. For example, UOS sediments do not increase DNF rates in response to elevated $NO_x$ concentrations to the same degree as other habitats, like swamp forest, do (Figure 2). Therefore, under high nitrate conditions, UOS contribute a smaller proportion to nitrogen removal than under low nitrate conditions (Figure 6). The same is observed with subtidal sediments. Swamp forests, on the other hand, reliably increase the proportion of their contribution under high nitrate conditions.






**Table 2.** Summary of $NO_x$ concentrations (μM) and average nitrogen loads ($NO_x$ and TDN; kg h$^{-1}$) on the day of maximum discharge, and percent of load removed by habitats under low and high nitrate conditions. Asterisks indicate the more representative percentage based on water column nitrate concentrations during different storm types.

| Storm | Type | [$NO_x$] | $NO_x$ load | TDN load | % TDN load removed- Low nitrate | % TDN load removed- High nitrate | % $NO_x$ load removed- Low nitrate | % $NO_x$ load removed- High nitrate |
|---|---|---|---|---|---|---|---|---|
| Joaquin | HP-HW | 21.1 | 295 | 898 | 3.05* | 12.8 | 9.27* | 38.8 |
| Matthew | HP-HW | 8.50 | 597 | 3060 | 1.15* | 4.81 | 5.90* | 24.6 |
| Florence | HP-HW | 5.33 | 303 | 2730 | 2.40* | 9.40 | 21.6* | 84.6 |
| Charley | HP-LW | 17.3 | 212 | 659 | 3.31 | 13.3* | 10.3 | 41.3* |
| Dorian | HP-LW | 22.7 | 350 | 1010 | 4.01 | 16.5* | 11.6 | 47.7* |
| Arthur | LP-HW | 36.0 | 218 | 431 | 5.95 | 24.6* | 11.8 | 48.7* |
| Hermine | LP-HW | 24.8 | 268 | 726 | 2.11 | 8.05* | 5.71 | 21.8* |


## 4 Discussion

The results of this study shed light on nitrogen removal capacities of multiple flood-prone natural and human-influenced habitats. Few studies have investigated nitrogen removal by habitats in the context of a built environment (Denman et al., 2016; Groffman & Crawford, 2003; Reisinger et al., 2016; Rosenzweig et al., 2018), and even fewer studies have quantified

nitrogen processing by urban landscapes, such as stormwater ponds and lawns (Gold et al., 2017; Hohman et al., 2021; Raciti et al., 2011). As coastlines continue urbanizing, these features will become increasingly abundant and comprise an important piece of coastal nutrient budgets. Additionally, this study explores how characteristics of storms can influence nitrogen removal capacity by coastal landscapes. This information is important in the context of climate change and the projected increase in the rainiest storms.

**4.1 Nitrogen removal by habitat treatments**

Positive $N_2$-N flux rates indicate that all habitats are capable of permanently removing nitrogen under high and low nitrate conditions, although, DNF rates varied across habitats and between nitrate treatments for some habitats. Under low nitrate conditions, marsh sediments produced the highest DNF rates and shallow subtidal sediments produced the lowest. Differences in DNF rates between structured habitats, like marshes, and unstructured habitats, like subtidal sediments have been

documented in previous studies (Piehler & Smyth, 2011). These differences have been attributed predominantly to the availability of organic carbon. Carbon availability may explain differences between marshes and other habitats as well. Swamp forests are structured habitats, like marshes; however, the forest sediments produced DNF rates that were lower than the marsh





sediments under low nitrate conditions. It is possible that the quality of carbon affects DNF (Hill & Cardaci, 2004; Seitzinger, 1994). The organic carbon supplied to the sediments by forests, could be more recalcitrant than that of marshes. In subtropical

systems, it has been shown that the molecular structure of sediment organic carbon in marshes are simpler and more readily decomposed than their woody counterparts, mangroves, which were dominated by recalcitrant forms of carbon (Sun et al., 2019).

There was not a significant difference between DNF rates in marshes and UOS under low nitrate conditions. The few studies

that have examined nitrogen processing in urban UOS, have shown that grasses can exhibit high DNF activity (Groffman et al., 1991), but is spatially and temporally heterogenous (McPhillips et al., 2016; Raciti et al., 2011). Multiple factors can influence nitrogen processing, including fertilizer application, soil saturation, species of grass, and temperature (Mancino et al., 1988; Wang et al., 2013). A 1998 study (Mancino et al.) examined nitrogen dynamics in turfgrasses under multiple temperature and soil saturation conditions, and found that when soils are saturated and water temperatures are high, grasses

can remove up to 93 % of applied nitrogen via DNF. Experimental conditions in this NRE study are similar to those in the 1998 study—saturated soils and warm temperatures—and results suggest that OUS makes up an important piece of this system's nitrogen budget under low nitrate conditions.

UOS sediments were unique in that they were the only habitat that exhibited 100 % DNF efficiency under low nitrate

conditions; all nitrogen that fluxed out of the sediments was in the form of $N_2$-N. This could be explained, in part, by organic carbon availability. Eyre and Ferguson (2009) describe critical carbon loading range to maximize DNF efficiency. It is possible that the high DNF efficiency exhibited by UOS sediments under low nitrate conditions is due to the presence of labile organic carbon in the soils that falls within a critical range; extremely high organic carbon may create a thick anoxic layer that is unsuitable for aerobic nitrifying bacteria that produce nitrate used in DNF. In contrast, too low organic carbon may promote

an aerobic layer unsuitable for the anaerobic denitrifying bacteria. Additionally, grasses have been shown to be extremely efficient at nutrient assimilation (Petrovic, 1990). It is plausible that much of the inorganic nitrogen that would have otherwise fluxed out of the sediments was integrated into biomass.

Like grasses, stormwater ponds are prolific features of developed landscapes, and yet few studies have examined their nitrogen

processing capabilities. Low nitrate-DNF rates exhibited by the stormwater pond sampled in our study were low relative to the marsh but were higher than those from other studies (Gold et al., 2017; Gold et al., 2021). The pond sampled in this study is part of a restored wetland that feeds a tributary creek of the Neuse River. High DNF rates in this pond relative to others suggest that hydrological design as well as ecological connectivity could increase nitrogen removal by stormwater ponds. Flooding from the Neuse River could increase circulation to reduce stratification and prolonged anoxia that appears to favor alternate

nitrogen pathways. Additionally, the natural vegetation that surrounds the stormwater pond could provide a source of organic carbon to the sediments, much higher than a typical urban stormwater pond (Blaszczak et al., 2018; Hohman et al., 2021).



Following nitrate enrichment, all habitats exhibited significant increases in DNF. This type of biogeochemical response has been observed in other studies (Gold et al., 2021; Seitzinger, 1994; Smyth et al., 2015); however, DNF rates under the high

nitrate condition in this study were exceptionally high, with marsh, swamp forest, and stormwater pond sediments producing the highest rates. These habitats also showcased the highest percentages of SOM. It is possible that these high SOM environments were nitrate limited and organic carbon was in excess; therefore, under low nitrate conditions there is a portion of SOM that was not used in the DNF process. This is supported by the weak linear relationship between SOM and DNF under low nitrate conditions. The significant positive linear relationship between SOM and DNF under high nitrate conditions

supports that an external source of nitrate may have alleviated this limitation with abundant SOM readily available. A similar phenomenon was observed in Arango et al.'s (2007) study examining denitrification in streams in the Midwest of the US. To summarize, habitats showed increased DNF capacity in response to elevated $NO_x$ concentrations, with high SOM environments playing the most important roles in nitrogen removal when water column $NO_x$ concentrations are high.

The significant positive linear relationship between DNF and SOD at low nitrate concentrations is consistent with results obtained by Piehler and Smyth (2011) and Seitzinger et al. (2006). This suggests that DNF is tightly coupled with nitrification; the nitrate fueling DNF is produced *in situ*. Under high nitrate conditions, the relationship between SOD and DNF is no longer significant. This weak relationship paired with negative $NO_x$ fluxes post-nitrate enrichment, could point to an increased importance of direct DNF, where overlying water supplies nitrate for DNF in the sediments.

**4.2 Nitrogen removal during different types of storms**

Comparing multiple storms that have affected North Carolina's coast revealed that HP-HW storms result in water column $NO_x$ concentrations that are significantly lower than HP-LW, LP-HW, and LP-LW storms. It is possible that during HP-HW events, there is a compound dilution effect from both riverine discharge and wind-driven storm surge. $NO_x$ concentrations can significantly alter biogeochemical processes, namely DNF. Therefore, it is likely that the effectiveness of nitrogen removal by

the coastal landscape is dependent on the type of storm impacting the region. Results from this work suggest that flooded landscapes are permanently removing water column nitrogen through direct DNF at higher rates during HP-LW, LP-HW, and LP-LW storms compared to HP-HW storms when $NO_x$ concentrations are relatively low and coupled nitrification-DNF is likely the dominant process.

These results largely reinforce the idea put forth by Paerl et al. (2018) where the Neuse River acts as a 'pipeline', delivering nitrogen to Pamlico Sound during these rainier events, as opposed to a 'processor' during drier events. Though $NO_x$ concentrations during HP-HW storms were relatively low, the high volume of water during wetter storms delivers larger loads of TDN and nitrate to the estuary compared to drier storms. Reduced nitrogen removal capacity of the coastal landscape during these flood events paired with increases in nitrogen loads has implications for greater nitrogen export into Pamlico Sound,

especially as climate changes increase the magnitude and frequency of these rainier storms (Easterling et al., 2017; Knutson et al., 2010; Paerl et al., 2019).

Potentially exacerbating this threat to water quality is development within the Neuse River watershed. This study sheds light on how human impacts on the landscape influence distributions of nitrogen sinks as anthropogenic nitrogen sources increase.

As urban and suburban landscapes expand, UOS will become more abundant and their role in regulating water quality will grow. These results suggest that under low nitrate conditions, DNF rates in UOS sediments are comparable to marshes and they will play an important role during flooding from HP-HW storms, and other low nitrate scenarios. They likely will not play as large a role during other types of storms when water column nitrate concentrations are relatively high.

Just as developed landscapes are expanding within watersheds and along coastlines, it seems as though floodplains are infringing on these built environments (Sebastian et al., 2019; Wang et al., 2017; Wobus et al., 2017). Some experts cite a regime shift in tropical cyclone activity where annual exceedance percentages historically used to delineate floodplains (e.g., 100-year and 500-year floodplains) are no longer representative of the current conditions (Paerl et al., 2019). When assessing landcover within each storm's floodplain and the nitrogen removal by each habitat, UOS played an increasingly important role

during the most recent storms (Matthew, Florence, and Dorian). It is possible that their growing abundance within storm floodplains and their increased contribution to nitrogen removal informs a trend where floodplain boundaries are encroaching further inland (Knutson et al., 2013; Min et al., 2011).

In the upper NRE, when $NO_x$ concentrations are high, more natural landscapes—including the hydrologically and ecologically

connected stormwater pond sampled in this study—produced the highest DNF rates. However, the limited areal extent of marshes and stormwater ponds within each storm's floodplain rendered them incapable of removing substantial nitrogen in this region. Swamp forests, on the other hand, appear to play an important role in maintaining water quality during storms. They were consistently the most extensive habitat within the storms' floodplains and, as such, made the largest contribution to nitrogen removal under both low and high nitrate conditions. Therefore, swamp forests appear to be essential for regulating

water quality in the NRE during storms of varying characteristics.

**5 Conclusions**

The results of this study provide information about nitrogen removal capacities by multiple natural habitats and urban landscape features in a flood-prone, developed, estuarine environment. All habitats showcased the ability to permanently remove nitrogen from the system under low nitrate conditions and increase nitrogen removal capacity in response to additional

nutrients delivered to the system. Flooded UOS can play an important role in regulating nitrogen when water column concentrations are relatively low (e.g., during rainier & windier storms). When water column nitrate concentrations are high,



more "natural" structured habitats, including marshes and swamp forest along with a somewhat unique stormwater/wetland pond, were more effective at removing nitrogen than shallow subtidal sediments and UOS. These differences in processing suggest that abundance and spatial distributions of these habitats within a floodplain can influence overall nitrogen removal
capacity at the watershed scale.

Water column nutrient concentrations produced by different types of storms likely influence biogeochemical processing by flooded habitats and subsequent nitrogen export downstream and into Pamlico Sound. Results of this study suggest that flooded landscapes may be less effective at removing reactive nitrogen during HP-HW storms compared to other storm types. Low
water column $NO_x$ concentrations produced during HP-HW events, likely result in relatively low DNF rates. DNF rates are likely higher during storm events that produce relatively high water column $NO_x$ concentrations, such as HP-LW, LP-HW, and LP-LW storms. Swamp forests are the most extensive of the habitats in this study area and play an important role in removing nitrogen and regulating water quality, regardless of storm characteristics. Management strategies should continue prioritizing swamp forest conservation in this region, as in North Carolina's Riparian Buffer Protection Program.


As coastlines and watersheds become more developed and climate change increases the frequency and magnitude of storms and major flooding events, the export of both anthropogenic and terrigenous nitrogen will likely increase. Understanding nitrogen removal capabilities and limitations of flooded natural coastal habitats as well as those urban landscapes that will become more and more prevalent, will enable us to make informed management decisions to benefit the integrity of our coastal
waters.

**Data availability**

Land cover data, watershed boundaries, and flood footprint data are open source and accessible online
(https://www.mrlc.gov/data; https://www.usgs.gov/national-hydrography/watershed-boundary-dataset;
https://cera.coastalrisk.live/). Wind and riverine discharge datasets are also open source and accessible online
(https://www.ndbc.noaa.gov/station_page.php?station=clkn7; https://waterdata.usgs.gov/monitoring-location/02091814/).
Data collected during the sediment core incubation experiment are available upon request.

**Acknowledgements**

This research was supported by funding from The National Science Foundation Growing Convergent Research Award #2021086 and the North Carolina Collaboratory award. We would like to thank Piehler lab members at the UNC Institute of
Marine Sciences, Mollie Yacano and Chelsea Brown, for their assistance with sample collection and laboratory experiments. We also thank the anonymous reviewers for their helpful comments.



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
