# Peer review of "Storm characteristics influence nitrogen removal in an urban estuarine environment"

_EGUsphere, 2023_

## Author Response (AR2)

We appreciate the thoughtful feedback provided by Anonymous Referees 1 and 2. Addressing these comments will improve the quality of this manuscript. Below, I have addressed each comment.

**Response to Anonymous Referee #1:**

Comment (summary statement): It is important that the authors acknowledge that their measurements include and N fixation (and maybe annamox, too) occurring simultaneously in sediments; as such, their measurements aren't really estimating 'denitrification' specifically.

Response: Thank you for making this important clarification. We mention in the methods section (line 157) that we are measuring net denitrification but will be more detailed in our explanation of processes that affect net denitrification, including nitrogen fixation and potentially annamox.

Comment (line 9): Suggest changing to "...(including NOx)...", since floodwaters bring additional reactive N forms (NH4, urea) that could be subjected to denitrification eventually.

Response: Thank you, this change has been made.

Comment (lines 14-15): Suggest changing to "Continuous-flow sediment core incubation experiments were used to quantify...", since 'flow-through' implies that the flow was through the sediments (bottom to top or v.v.), and the incubations themselves aren't doing the quantifying.

Response: Thank you for making this clarifying point, these changes have been made.

Comment (line 43): Here and elsewhere, suggest avoiding wording like "has been shown to exacerbate"...instead, consider "Watershed urbanization exacerbates...". Line 52, consider "Some natural habitats are effective at removing...". Etc.

Response: Thank you for the reminder. Changes have been made throughout the manuscript.

Comment (lines 58, 60): Consider changing "by" to "in" (microbes within these environments are performing DNF, not the habitats or landscapes).

Response: Thank you, this change has been made.

Comment (line 61): Is "prolific" the right word here? Perhaps "...despite being common...", or maybe "ubiquitous"?

Response: Thank you, this change has been made.

Comment (line 62): Suggest changing "is" to "was" (and also line 75), and "denitrification" to "DNF". Response: Thank you, these changes have been made.

Comment (line 67): Consider deleting "Studies have shown that".

Response: Thank you, this change has been made.

Comment (line 81): Consider editing to "...will enable estimation of landscape-scale..."

Response: Thank you, this change has been made.

Comment (line 86): Suggest changing "combines" to "combined".

Response: Thank you, this change has been made.

Comment (lines 117-118): Suggest changing to "Furthermore, storms classified as LP-LW were considered "baseline storm" conditions."

Response: Thank you, this change has been made.

Comment (line 130): Suggest changing to "This procedure was necessary..." (or similar word).

Response: Thank you, this change has been made.

Comment (line 139): Suggest changing "mean flux rates" to "mean fluxes" here and throughout the paper (fluxes are already rates, so "flux rates" is redundant).

Response: Thank you, this change has been made throughout the manuscript.

Comment (line 147): If I understand correctly, the inflow water does not actually go 'through' the sediment cores, but rather into the overlying water column (seems to be confirmed by Piehler & Smyth 2011). In that case, suggest using "continuous-flow" instead of "flow-through".

Response: Thank you for making this clarifying point. That is correct that the inflow water is cycled through the overlying water column, not the sediments. The nomenclature has been adjusted throughout the manuscript.

Comment (line 156): Does "water pulled from cores" = "outflow water"? If so, the latter terminology might avoid confusion, since "pulled from cores" could mean sampling directly from the overlying water using a sampling port or syringe, etc. (some readers might miss that the peristaltic pump 'pulls' in your case).

Response: Thank you for making this point. Yes, "outflow water" is synonymous to "water pulled from cores". Changes have been made to improve clarity.

Comment (line 157): Suggest explaining why it's "net DNF" (presumably as opposed to net N fixation). What is actually being measured is the amount by which DNF (plus any anammox) exceeds simultaneous N fixation (if any).

Response: Thank you for bringing up this important point. The text has been updated to clarify the multiple processes that can affect N2 fluxes, including DNF, nitrogen fixation, and annamox.

Comment (line 160): "Quikchem". For future studies, note that 0.7  $\mu$ m filtration can lead to large fluctuations in measured concentrations, especially for NH4 (presumably from bacteria/archaea, etc., passing through the filter; see Reed et al. 2023; L&O: Methods 21 (1): 1-12).

Response: Thank you for this valuable suggestion. We will consider this important point during future experiments.

Comment (line 195): "multiply by 100".

Response: Thank you, this change has been made.

Comment (line 205): Delete extra "were compared". Presumably, TDN load still includes the NOx part of the load? Might be good to clarify this point for the reader.

Response: Thank you for these comments. In the methods section, we describe our nutrient analysis methods and explain that dissolved NOx, NH4, and organic nitrogen are components of TDN. However, as you suggest, it would be helpful to the readers to reiterate that point here. This change has been made.

Comment (line 221): Suggest mentioning on first use what the variation term is (std dev, std error, etc.). The number of observations (n) would also be useful.

Response: Thank you for these suggestions, changes have been made.

Comment (Fig. 2): Caption needs additional explanation (i.e., lower-case letters) to allow it to stand alone.

Response: Thank you for pointing this out. The compact letter display is meant to denote statistical significance, which is not clear in the text. We have edited this figure and caption to improve clarity.

Comment (lines 240-241): Suggest changing "hovered" to "was" and deleting "evident".

Response: Thank you, these changes have been made.

Comment (line 242): Delete "nitrate".

Response: Thank you, this change has been made.

Comment (Fig. 3): To clarify, are the units for DNF DNF  $\mu$ mol N m-2 h-1 or  $\mu$ mol N2 m-2 h-1? NOx influxes (at high NO3) seem to be consistently much lower than N2 effluxes in all systems (assuming that the units for each allow direct comparison), which suggests that denitrifiers are still getting a lot of NOx from other sources (coupled to nitrification probably, especially since NH4 fluxes are relatively low). The paragraph starting at line 375 seems to downplay the continuing importance of coupling at high NO3. Sure, at high NO3, direct denitrification is going to be more important than it is at low NO3, but the coupling still seems to be a lot more important than direct (the different y-axis scales are masking this pattern).

Response: Thank you for making these important points. The units for denitrification are  $\mu$ mol N m-2 h-1. Through the manuscript, this is written as  $\mu$ mol N2-N m-2 h-1, following the format used in papers such as Piehler & Smyth et al., 2011 and Yacano et al., 2022. Furthermore, our intention was not to reduce the role of coupled nitrification-denitrification under high nitrate conditions, but rather emphasize its relative importance during low nitrate conditions while highlighting the increased role of direct denitrification under high nitrate conditions. However, it is critical that we make this clarification, and we have adjusted the text as such.

Comment (line 274): Suggest changing first "of" to "for".

Response: Thank you for this comment. Changes have been made.

Comment (line 293): Suggest changing "are" to "were".

Response: Thank you, this change has been made.

Comment (line 295): Suggest changing "that reliably" to "and".

Response: Thank you, this change has been made.

Comment (lines 300-303): Suggest using past tense (do --> did; contribute --> contributed; is --> was; increase --> increased), and changing "like swamp forests, do" to "such as swamp forests".

Response: Thank you for these suggestions. Changes have been made.

Comment (line 316): Consider changing "will become" to "are".

Response: Thank you, this change has been made.

Comment (lines 318-319): Sentence ends a bit awkwardly. Perhaps "...increase in high precipitation storms." instead?

Response: Thank you for this suggestion. Your proposed change has improved the flow of this sentence.

Comment (line 327): Consider "exhibited" rather than "produced".

Response: Thank you, this change will be made.

Comment (lines 328-332): Suggest editing to "The quality of organic matter affects DNF (add citations), and the molecular structure of sediment organic matter in marshes is simpler and more readily decomposed than sediment organic matter in mangrove forests (Sun et al., 2019). Thus, sediment organic matter in the inundated forests in the present study may have been more recalcitrant than the organic matter in the marshes."

Response: Thank you for the suggestion. These changes have been made to improve the structure and clarity of this paragraph.

Comment (lines 338-342): Suggest editing to "When soils are saturated and temperatures are high, turfgrass sediments can remove up to 93% of applied nitrogen via DNF (Mancino et al., 1998). Experimental conditions in the present study were similar to those in Mancino et al., and results suggest that UOS are important for system nitrogen budgets under low nitrate conditions."

Response: Thank you for the suggestion. These changes have been made to improve the structure and clarity of this paragraph.

Comment (lines 345-346): This sentence could perhaps be deleted to improve flow.

Response: Thank you for this comment, changes have been made.

Comment (lines 346-347): Consider deleting "It is possible that", and change "is" to "may be".

Response: Thank you for these suggestions, changes have been made.

**Comment (lines 357-360): A bit confusing what the relevance of these sentences are for the study. What is it about this particular hydrological design that contributed to high DNF? Which alternative nitrogen pathways, and why are they (presumably) less desirable than DNF?**

Response: We appreciate this observation. The reference we cited (Gold et al., 2017) describes these alternative processes, that are likely stimulated due to poor circulation, thermal stratification, and anoxia in the bottom waters of traditional stormwater ponds. Alternative, less favorable processes include DNRA that could increase the supply of inorganic nitrogen to the system, as well as increased phosphate flux from the sediment to the bottom waters resulting in nitrogen limitation. In tandem, these processes may trigger algal blooms that degrade water quality. In the manuscript, we suggest that connectivity to the Neuse River could improve circulation and mitigate conditions that could degrade water quality. We have adjusted the text to explain these processes in further detail.

**Comment (lines 365-366): 'exceptionally high' compared to what? Suggest "exhibited" instead of "showcased".**

Response: Thank you for bringing up these points. Changes have been made according to your suggestions. Initially, the comparison was to DNF rates measured in other urban coastal systems (e.g., Rosenzweig et al., 2018). However, this comparison may not be appropriate as those studies were conducted in sites with considerably different physical characteristics (e.g., high salinity, tidal influence). Furthermore, the high nitrate scenario and habitat treatments are unique to the present study. Thus, the portion of the sentence "... however, DNF rates under the high nitrate condition in this study were exceptionally high..." has been deleted.

**Comment (line 373): Suggest changing "are" to "were".**

Response: Thank you, this change has been made.

Comment (line 376): Suggest editing to "...2006), suggesting that DNF was tightly coupled...".

Response: Thank you, this change has been made.

Comment (lines 377-378): Suggest changing "is" (both of them) to "was", and editing to "This weak relationship, as well as negative NOx fluxes,...".

Response: Thank you for these suggestions, these changes have been made.

Comment (lines 382-383): Suggest changing "are" to "were", and editing to "During HP-HW storms, riverine discharges and wind-driven storm surges may dilute NOx concentrations."

Response: Thank you for these suggestions, these changes have been made.

Comment (lines 384-385): Suggest deleting "it is likely that" and editing to "...landscape may depend on...".

Response: Thank you for these suggestions, these changes have been made.

Comment (line 378): Suggest adding commas before and after "compared to HP-HW storms".

Response: Thank you, this change has been made.

Comment (line 393): Were there any patterns in terms of TDN composition for the various storms? I.e., did NOx comprise a larger proportion of TDN for some storm types vs. others? Here is where it would be good to know for sure that the TDN still includes the NOx part (as I've assumed it does).

Response: Thank you for this thoughtful comment, we have adjusted the text to clarify that TDN includes dissolved NOx, dissolved NH4, and dissolved organic nitrogen. The proportion of nitrogen species was not something we had previously considered. However, after reading your comment, we determined that the proportion of NOx is higher during low precipitation storms (74.16% in LP-HW storms and 44.78% in LP-LW storms) than it is during high precipitation storms (16.49% in HP-HW storms and 20.81% in HP-LW storms). It is possible that this difference can be attributed to the high amounts of "other things", like terrigenous organic nitrogen, that enters the system during high precipitation events, lowering the proportion of nitrate. Furthermore, with increased volume due to high amounts of precipitation, there is likely that dilution decreases the proportion of nitrate under the highest flows. We appreciate your feedback which prompted this interesting finding. We have included this in the revised manuscript.

Comment (lines 423-425): Suggest editing to "All habitats removed nitrogen under low nitrate conditions and increased their nitrogen removal capacity in response to added nitrate."

Response: Thank you for this suggestion, this change has been made.

Response to Anonymous Referee #2:

Comment (summary statement): ... I'm wondering why the experimental levels chosen (16.8 and 52.3 uM) were higher than those from the storms? This may have led to an over estimate of the DNF rates and should at least be discussed.

Response: Thank you for raising this important question. NOx concentrations in the low nitrate scenario were determined by ambient conditions at the time of sampling, which were unknown until we conducted the experiment. The high nitrate treatment involves the addition of sodium nitrate to the feedwater mid-experiment and is intended to replicate the high NOx concentration during some storms. While the high NOx concentrations were higher than the means calculated for HP-LW, LP-HW, and LP-LW storms, they were very close to the highest NOx concentrations calculated in our weighted regression analysis. Of the storms we assessed, Hurricane Arthur, an LP-HW storm, produced the highest average NOx concentration at 47.6 uM. We are careful to label these treatments "more representative" or "more analogous" to conditions during specific storms (e.g., line 269). However, as you suggest, the potential overestimation bares more discussion, and we have adjusted the text as such.

Comment (summary statement): This suggests that addition of nitrate either stimulated nitrification or that anammox was occurring. The authors discuss some of this in the section beginning on line 375 but I think it could be expanded a bit, and annamox needs to be mentioned.

Response: Thank you for this comment. You are correct in that the increase in denitrification rates are greater than NOx fluxes into the sediments under high nitrate conditions. As you suggest, it is important to mention other processes that potentially contribute to the increase in denitrification. We discussed the importance of coupled nitrification-denitrification under low nitrate conditions, but regretfully did not convey the continued importance of this process under high nitrate conditions. We have revised the text to clarify this point and to discuss the potential role of annamox in elevating denitrification rates.

**Comment (lines 39-41): This sentence seems a bit out of place**

Response: Thank you for this helpful comment. The intent of this paragraph is to convey how floodwaters can stimulate water quality degradation by triggering algal blooms and subsequent anoxia. Thus, we think this sentence is important to include. However, we have moved it to earlier in the paragraph to improve flow.

Comment (line 61): Perhaps better to say despite them making up a significant amount of the area in developed settings.

Response: Thank you for this suggestion, this change has been made.

Comment (line 74): Secondary objective implies a less important objective. It seems this is more of an application of the information gathered from the first objective.

Response: Thank you for making this important clarification. We do not want to suggest that one objective is more important than the other. Thus, we have changed the phrasing from "primary" and "secondary" objectives to "One objective of this study was…" and "An additional objective of this work was…". We have also changed the text to be more explicit about the application of data collected as part of the previous objective.

**Comment (line 215): why were storm water ponds eliminated, was there no data on their area?**

Response: We appreciate this important question. Landscape-scale nitrogen removal was calculated using National Land Cover Datasets (NLCD) that coincided with hurricane landfall dates. NLCD files are 30-meter rasters, which are too coarse to accurately capture most of New Bern's stormwater ponds. The City of New Bern provided us with their stormwater infrastructure geospatial datasets, which included shapefiles of stormwater ponds within city limits but not within the two watersheds that were used in our analysis. Furthermore, information on pond age was unavailable. Due the coarse spatial resolution of the NLCD and the lack of temporal resolution of New Bern's stormwater data, we decided to exclude storm water ponds from this analysis.